# Which Antibiotic for Urinary Tract Infections in Pregnancy? A Literature Review of International Guidelines

**DOI:** 10.3390/jcm11237226

**Published:** 2022-12-05

**Authors:** Mariela Corrales, Elizabeth Corrales-Acosta, Juan Guillermo Corrales-Riveros

**Affiliations:** 1Department of Urology AP-HP, Tenon Hospital, F-75020 Paris, France; 2Obstetrics and Gynecology Service, Marina Baixa Hospital, 03570 Villajoyosa, Spain; 3Department of Urology, Clínica Ricardo Palma, Lima 15036, Peru

**Keywords:** urinary tract infection, UTI, pregnancy, woman, guideline, asymptomatic, bacteriuria, cystitis, pyelonephritis

## Abstract

Urinary tract infection (UTI) is considered to be a major problem in pregnant women. It is also one of the most prevalent infections during pregnancy, being diagnosed in as many as 50–60% of all gestations. Therefore, UTI treatment during pregnancy is extremely important and management guidelines have been published worldwide to assist physicians in selecting the right antibiotic for each patient, taking into account the maternal and fetal safety profile. A review of the literature was carried out and all international guidelines giving recommendations about antibiotic treatments for pregnancy-related UTI were selected. The search came back with 13 guidelines from 4 different continents (8 from Europe, 3 from South America, 1 from North America and 1 from Oceania). Our review demonstrated concordance between guidelines with regard to several aspects in the antibiotic treatment of UTI during pregnancy and in the follow-up after treatment. Nonetheless, there are some areas of discordance, as in the case of antenatal screening for bacteriuria and the use of fluoroquinolones in lower or upper UTI. Given the current evidence that we have from international guidelines, they all agree on several key points about antibiotic use.

## 1. Introduction

Urinary tract infection (UTI) is considered to be a major problem in pregnant women [1,2,3]. It is also one of the most prevalent infections during pregnancy, being diagnosed in as many as 50–60% of all gestations [4].

UTIs can be classified as lower urinary tract infections, including both asymptomatic bacteriuria (ASB) or acute cystitis (AC), and upper urinary tract infections or acute pyelonephritis (APN) [5]. Most infections are caused by *Enterobacteriaceae*, commonly found in the gastrointestinal tract, with *Escherichia coli* (*E. coli*) being responsible for 80–90% of cases. However, we can find other bacteria such as *Group-B Streptococcus saprophyticus* (GBSS), *Klebsiella pneumoniae*, *coagulase-negative Staphylococcus*, *Staphylococcus aureus* and *Proteus mirabilis* in a lower percentage [2,6].

In pregnant women, ASB occurs in an estimated 2–10% [7], and if left untreated, it can turn into symptomatic AC in 30% of patients and may progress to APN in up to 50% of those patients [6], which have been associated with several complications for both the mother and the unborn child [2,8].

Therefore, UTI treatment during pregnancy is extremely important and management guidelines have been published worldwide to assist physicians in selecting the right antibiotic for each patient, taking into account the maternal and fetal safety profile [5,6,9].

The aim of this study is to review the concordance in recommendations between evidence-based guidelines for antibiotic treatment of pregnancy-related UTI developed by different authorities around the world. Additionally, we will review their concordance in terms of ASB screening and follow-up after treatment.

## 2. Methods

A literature review was carried out in August 2021 using the PubMed and Scopus databases for clinical guidelines covering the topic of pregnancy-related UTI. An additional search was performed in the Guidelines International Network (G-I-N) for any relevant guidelines not identified by our PubMed and Scopus database search. Exclusion criteria included guidelines that did not include recommendations about antibiotic treatments for pregnancy-related UTI. This review followed the Preferred Reporting Items for Systematic Reviews and Meta-Analyses (PRISMA) statement [10].

Different searches were carried out with the following medical subject heading (MeSH) terms and keywords: “urinary tract infection”, “UTI”, “pregnancy”, “woman”, “guideline”, “asymptomatic”, “bacteriuria”, “cystitis” and “pyelonephritis”. Boolean operators (AND, OR) were used to refine the search. The references of each included guideline were also reviewed. No time period nor language restrictions were applied.

## 3. Results

The PubMed and Scopus search returned 386 results and 20 additional guidelines were added after the G-I-N search. After duplicate removal and review of results, a total of 20 guidelines were selected of which 7 were excluded leaving 13 guidelines that fulfilled our inclusion criteria. The summary of the selection process is represented in Figure 1.

Of the 13 guidelines coming from 4 different continents (Table 1), 8 came from Europe, produced on behalf of the European Association of Urology (EAU) [7], German Society of Urology (German acronym: DGU) [11], Swiss Society of Gynaecology and Obstetrics (SSGO) [12], Spanish Society of Clinical Microbiology and Infectious Diseases (SEIMC) [13], joint report of French Infectious Diseases Society/Urological French Association (French acronym: SPILF/AFU) [14], joint report of the Institute of Obstetricians and Gynaecologists, Royal College of Physicians of Ireland/Clinical Strategy and Programs Division, Health Service Executive (IOGRCPI/ CSPDHSE) [15] and 2 guidelines coming from the National Institute for Health and Care Excellence (NICE) [16,17]; 1 from North America, produced on behalf of the Infectious Diseases Society of America (IDSA) [18]; 3 from South America, produced on behalf of the joint report of the Brazilian Society of Infectious Diseases/Brazilian Federation of Gynecology and Obstetrics Associations/Brazilian Society of Urology/Brazilian Society of Clinical Pathology/Laboratory Medicine (SBI/FEBRASGO/SBU/SBPC/ML) [19], joint report of the Argentinean Society of Infectious Disease (SADI) [20] and Colombian Association of Infectious Disease (Spanish acronym: ACIN) [21]; and 1 from Oceania, produced on behalf of South Australian Health (SAH) [22].

For better understanding of the main purpose of the present report, the obtained results have been divided into different sections. Those sections are organized as follows: screening for ABU, antibiotics in ABU, antibiotics in cystitis, antibiotics in APN, urine culture follow-up and prophylaxis follow-up.

Key points of antibiotic use in pregnancy according to international guidelines (for ASB, cystitis, APN and prophylaxis) are summarized in Table 2.

### 3.1. Screening for ABU

A total of 11 out of 13 guidelines (85%) recommended systematic screening for ABU bacteriuria in pregnant women. All guidelines coming from South America [19,20,21], North America [18] and Oceania [22] recommended this, as well as almost all guidelines coming from Europe [7,13,14,15,16], with the exception of Germany and Switzerland [11,12].

Most guidelines recommended to conduct this screening by a UC [7,13,14,15,16,18,19,20,21,22] at the first antenatal visit, ideally at 12–16 weeks and not later than 16 weeks [13,15,18,19,20,21,22]. Only French guidelines [14] recommended a monthly ABU screening at the fourth month of pregnancy with a UC or a urine test strip, except for patients at high risk of UTI, for whom a UC must be performed [14]. Three guidelines from Europe [7,15,16], one from North America [18], one from South America [19] and one from Oceania [22] agreed on taking the midstream specimen of urine (MSSU) for UC.

Almost all guidelines defined ASB as a urine sample showing ≥10^5^ colony-forming units (CFU)/mL without symptoms of UTI [7,8,9,10,11,12,13,14,15,16,19,20,21,22], ideally in two consecutive urine cultures [7,13,20,21]. However, for practical reasons, it is admitted that only one UC is enough [14,15,16,22].

Colombia and Brazil recommended to repeat this screening in the third trimester of pregnancy [19,21], especially in patients with chronic kidney disease (CKD), diabetes mellitus (DM) and history of UTI [21]. Argentina recommended to repeat this screening every 3 months with the presence of risk factors [22].

### 3.2. Antibiotics in ABU

Without specifying the hierarchy of preference, the recommended antibiotics were: nitrofurantoin [11,15,18,19,20], trimethoprim/sulfamethoxazole (TMP/SMX) [20], fosfomycin [11,12,13,15,19], multiple penicillins such as amoxicillin [15,19,20], ampicillin [18], pivmecillinam [11], ampicillin/sulbactam [21], amoxicillin/clavulanate [15,20,21], first-generation cephalosporine (1stGC) such as cephalexin [15,18,19,20,21] and second-generation cephalosporine (2ndGC) such as cefuroxime [15,19].

When referring to lines of treatment, as first line, nitrofurantoin [16,21], fosfomycin [21] and amoxicillin [14] were proposed. Second-line treatments were pivmecillinam [14], cephalexin [16] and amoxicillin (if sensible) [16]. France was the only country that gives up to five-line treatments [14], with fosfomycin being the third-line treatment; trimethoprim, the fourth-line one; and nitrofurantoin, amoxicillin/clavulanate, cefixime and TMP/SMX, the fifth-line treatment. TMP/SMX was also suggested as the last antibiotic choice in Germany [11].

Results are summarized in Table 3, including the dosage recommended by each guideline.

### 3.3. Antibiotics in Cystitis

Without specifying the hierarchy of preference, the recommended antibiotics were: nitrofurantoin [11,15,19,20], fosfomycin [7,11,13,15,16,19,20], multiple penicillins such as amoxicillin [15], pivmecillinam [11], amoxicillin/clavulanate [15,19,21], 1stGC (cephalexin) [15,20,21] and 2ndGC (cefuroxime) [15,19].

When referring to lines of treatment, as first line, nitrofurantoin [11,16,21], fosfomycin [14,21], trimethoprim [22], cephalexin [22], amoxicillin [16] and amoxicillin/clavulanate [12] were proposed. For second-line treatment the chosen antibiotics were pivmecillinam [14], cephalexin [16], penicillins such as amoxicillin [22] and amoxicillin/clavulanate [22] and 2ndGC (cefuroxime) [12]. For third-line treatment, options were nitrofurantoin [14], ciprofloxacin [14], TMP/SMX [12] and third-generation cephalosporine (3rdGC) such as cefixime [14]. TMP/SMX was also suggested as the last antibiotic choice in Germany [11].

Results are summarized in Table 3, including the dosage recommended by each guideline.

### 3.4. Antibiotics in APN

For APN treatment, most guidelines recommended as a first-line treatment a monotherapy with 3rdGC [13,14,15,20] such as ceftriaxone [15,20] or 2ndGC [7,13] such as cefuroxime [13,17] and, if patient unstable or septic, adding an aminoside such as gentamicin [7,13,15] was recommended. Double parental therapy with amoxicillin/gentamicin [7,22] or ampicillin/gentamicin [22] was also proposed.

Second-line treatment included cefuroxime [12], gentamicin [20], aztreonam [7,20], 2ndGC such as cefuroxime [12] and 3rdGC (ceftriaxone or cefotaxime) [22]. Ireland proposed a dual parenteral therapy with clindamycin or vancomycin (based on the susceptibly results) and gentamicin.

As a third-line option, France proposed ciprofloxacin in case of beta-lactamase allergy [14]. The AFU also specified a 10 day-treatment for UTI caused by extended-spectrum beta-lactamase producing *E. coli* (ESBLE). Ciprofloxacin, levofloxacin or TMP-SMX was the first-line choice, amoxicillin–clavulanate was the second-line choice and cefoxitin, piperacillin–tazobactam or temocillin was the third-line choice.

Recommendations for oral therapy switch were: amoxicillin [14,22], amoxicillin/clavulanate [14,22], cephalexin [22], trimethoprim [22], cefixime [14] or ciprofloxacin [14].

A couple of European guidelines proposed oral antibiotics for uncomplicated AP: cephalexin (1stGC) [17] or amoxicillin/clavulanate [12] as a first-line treatment and TMP/SMX [12] as third-line treatment.

Results are summarized in Table 4, including the dosage recommended by each guideline.

### 3.5. Urine Culture Follow-Up

After lower UTI treatment (ABU or cystitis), 8 out of 13 guidelines (62%) remarked the need of a UC follow-up after 7–14 days [11,13,14,15,19,20,21,22]. Additionally, Spanish speaking countries also recommended a monthly UC until delivery [13,14,20].

After treating APN, 4 out of 13 guidelines (31%) remarked the need for a UC after 7–14 days [11,14,20,22] and then monthly until delivery [14,20].

### 3.6. Prophylaxis Follow-Up

Prophylaxis was suggested in persistence of bacteriuria after full treatment [21], ≥2 episodes of UTI during pregnancy (ABU or cystitis) [19,22], after a single episode of UTI (ABU or cystitis) with history of UTI [19] or with risk factors for pyelonephritis [22]. The recommended antibiotics were cefalexin (250–500 mg) [19,21,22], nitrofurantoin (50–100 mg) [19,21,22] or fosfomycin (3 g single dose every 7–10 days) [21]. These antibiotics were suggested to be used in the postcoital regimen in those patients who have UTIs related to sexual activity [19,21], or continuously (at bedtime) [19,21,22]. If cephalexin is used, it must be stopped 4 weeks before delivery [21].

After one episode of APN, prophylaxis with nitrofurantoin was recommended by Irish guidelines [15]

## 4. Discussion

When choosing antimicrobials during pregnancy, safety considerations for both mother and fetus must be considered. Most of the antibiotics recommended by international guidelines are category B according to the United States Food and Drug Administration (FDA), meaning that there are no adverse effects in well-controlled studies of human pregnancies. However, some of the antibiotics used for UTI in pregnancy such as trimethoprim, TMP/SMX, gentamicin and ciprofloxacin are FDA category C, and must be used with caution [23].

Our review demonstrated concordance between guidelines with regard to several aspects in the antibiotic treatment of UTI during pregnancy and in the follow-up after treatment. Nonetheless, there are some areas of discordance, as in the case of antenatal screening for bacteriuria. There is adequate evidence showing that ASB is associated with an increased risk of APN, preterm labor and an increased risk of delivering a low-birth-weight infant, among other adverse fetal outcomes [8,24,25,26]. Moreover, studies have also shown a reduction of these complications by treating ASB in this population [27,28,29,30]. Based on this evidence we can describe a first scenario, screening for asymptomatic bacteriuria. All guidelines coming from North America, South America, the only one from Oceania and most European guidelines agreed on recommending systematic screening for ABU, even if most of the published studies have low-quality evidence. In summary, if we choose to carry out a screening for bacteriuria in pregnancy and we end up with a positive urine culture, then we can treat it tailoring the antibiotic treatment according to the weeks of pregnancy and urine culture sensitivity.

The only two European guidelines that do not recommend this screening anymore are the ones from Germany and Switzerland [11,12], with the exception of women at high risk for developing UTI (women with diabetes mellitus, immunosuppression, functional or structural abnormalities of the urinary tract, previous episodes of pyelonephritis, previous premature births or late pregnancy loss). This recommendation is mainly due to a recent high-quality study that demonstrated that in women with an uncomplicated singleton pregnancy, untreated ASB is related to a low risk of developing APN but it is not associated with an increased risk of premature birth or other neonatal or maternal complications [31].

Concerning antibiotic therapy for lower UTI, it was similar around the world. Few guidelines gave specific lines of treatments for ASB and cystitis. This may be related to variable patterns of antimicrobial resistance worldwide, meaning that treatment should be based on UC and sensitivities recommended by the laboratory report, taking into account the antibiotics allowed during the trimester of pregnancy. We remarked that only two European guidelines [12,16] and one from South America [21] highlighted the need for considering the local antimicrobial resistance profile (AMR) data when prescribing antibiotic treatment for UTI. Furthermore, German, Swiss and UK guidelines [11,12,16] were the only ones that made a statement about the AMR of each antibiotic according to their local population. A recent meta-analysis that investigated the AMR of different antibiotics used in pregnancy-related UTI, which also included studies from Europe and South America, showed that the most prevalent uropathogen was *E. coli*, followed by *Klebsiella* sp., two bacterial agents that were highly susceptible to nitrofurantoin [32]. This may be the reason why nitrofurantoin is still highly recommended by most international guidelines, being the first line of treatment in the UK and Colombia. Moreover, *E. coli* is also sensitive to 1stGC but extremely resistant to ampicillin and to other aminopenicillins worldwide [32,33]. Due to antimicrobial resistance, amoxicillin is not preferred as a first-line option, as specified by NICE and South Australian guidelines [16,22]. Instead, it can be recommended as a second-line treatment if there is no improvement of symptoms after using first-line antibiotics for at least 48 h, or when first-line treatment is not suitable [16].

The use of ciprofloxacin was another area of discordance in this review. French guidelines were the only ones that recommended ciprofloxacin as a third-line treatment for cystitis in pregnancy and as a first-line choice for ESBLE, together with levofloxacin or TMP-SMX [14]. In spite the fact that fluoroquinolones and TMP-SMX are both FDA category C, fluroquinolones are the only ones not reaching a consensus in international guidelines. Brazilian, Swiss and Irish guidelines do not recommend the use of ciprofloxacin during pregnancy [12,15,19], the latter being restricted to postpartum women only because of teratogenicity concerns [15]. Although fluoroquinolones have not been associated with increased risk of major malformations such as adverse effects in the musculoskeletal system, premature labor or intrauterine growth retardation [34,35,36,37], almost all guidelines do not mention them as an alternative treatment for UTI. In summary, this second scenario, which is treating a symptomatic UTI, recommends starting with an empiric antibiotic treatment according to the country’s guidelines.

In terms of therapy duration for lower UTI in pregnancy, all guidelines recommended the shortest possible duration, varying from 3 to 7 days for all antibiotics but fosfomycin. Recent publications show that there is no clear difference between a single dose vs. a 4–7 day short course of antibiotics for lower UTI treatment, in terms of progression to pyelonephritis (very low-quality evidence) and preterm birth (moderate-quality evidence) [23]. This would encourage the use of fosfomycin trometamol in patients with poor drug compliance, and also because this broad-spectrum bactericidal antibiotic has demonstrated excellent tolerability and safety in pregnancy [38]. However, Swiss guidelines still recommend a more prolonged therapy if there is increased risk of premature birth [12].

On the other hand, focusing on APN, it is well known that initial antimicrobial therapy is empiric and should be modified according to the UC results [5]. All guidelines agreed on giving lines of treatment for APN. Nonetheless, there was no consensus on the drug of choice for the first-line treatment. Empiric parenteral antibiotics included were 2ndGC or aminopenicillins (i.e., amoxicillin, ampicillin) associated with an aminoside (gentamicin) [7,13,14,15,16,20,22]. These traditional regimes have been associated with high efficacity withing the first 72 h [39,40,41] due to their ability to reach therapeutic concentrations in the upper urinary tract, contrary to nitrofurantoin and fosfomycin. However, concerns are being raised due to the AMR of the aminopenicillins [33]. Among international guidelines, Swiss and UK guidelines recommended starting with oral therapy, either with a 1st GC or amoxicillin/clavulanate, and as a second-line treatment parenteral antibiotic, if the patient is unable to take oral antibiotics or severely unwell [12,17]. All guidelines recommended switching to oral therapy, if the first antibiotic choice was parenteral, after 48 h of apyrexy [7,12,13,14,15,17,20,22]. These different schemes of treatment among guidelines suggest that all first lines of treatments proposed for ACP in pregnancy are similar in efficacity and should be used depending on the AMR. In fact, to date, there is no evidence that one treatment regimen for APN is better than another [39,40,41,42].

Guidelines from South America and Europe specified the need for a second UC 1–2 weeks after the antibiotic course has been completed. This is in line with previous recommendations [26,43]. Those same guidelines also suggest the need to follow a prophylactic treatment in certain cases, following the antibiotic cautions. Special management is needed for patients that have suffered from a UTI caused by GBSS, in whom prophylactic antibiotics are also needed during labor to prevent neonatal sepsis [44].

## 5. Limitations

The main strength of our review lies in the inclusion of guidelines from different continents, which can give us un idea about the worldwide management of pregnancy-related UTI. However, this study is not devoid of limitations. First, we acknowledge that some guidelines might have not been included because they were not found in our database search or because they vaguely mentioned the antibiotic treatment. Nonetheless, the aim of our study was to specifically present the antibiotic recommendations for UTIs. Second, the lack of specifications of certain guidelines in terms of dosage and optimal duration of antibiotic courses in pregnancy made the results more general.

## 6. Conclusions

Antibiotics selected for UTI during pregnancy should be safe for both mother and unborn child. Guidelines from the four selected continents agree on several key points about antibiotic use. First lines of treatment are similar for lower and upper UTI around the world; however, before selecting the antibiotic of choice, it is mandatory to know the AMR in the local population.

## Figures and Tables

**Figure 1 jcm-11-07226-f001:**
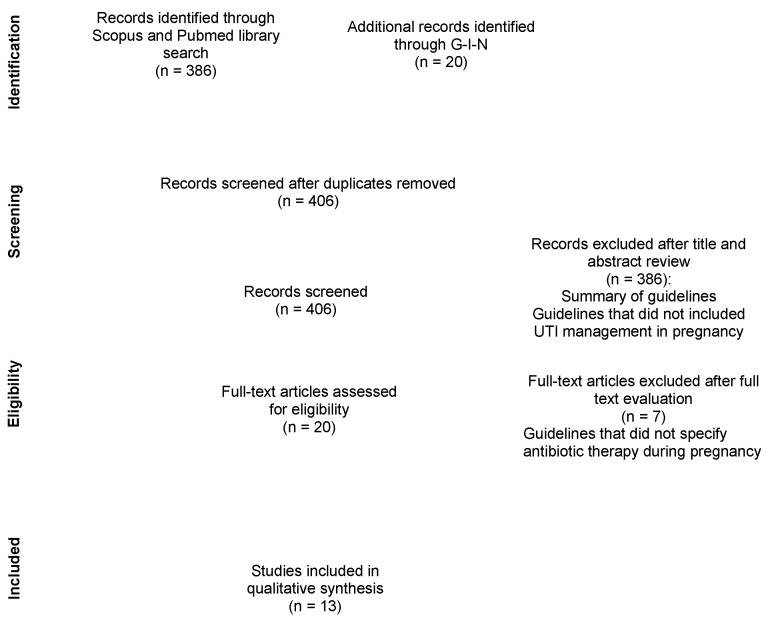
Flow chart of the literature review.

**Table 1 jcm-11-07226-t001:** International guidelines on urinary tract infections during pregnancy.

**North America**			
**Country/Region**	**Title**	**Organization**	**Year**
USA [18]	Clinical Practice Guideline for the Management of Asymptomatic Bacteriuria: 2019 Update	IDSA	2019
**Europe**			
**Country/Region**	**Title**	**Organization**	**Year**
EU [7]	Guidelines on urological infections	EAU	2020–update 2022
DE [11]	Interdisciplinary guide S3. Epidemiology, diagnosis, therapy, prevention, and management of community-acquired, bacterial, and uncomplicated urinary tract infections in adult patients	DGU	2011–update 2017
CH [12]	Guideline of the SSGO on acute and recurrent urinary tract infections in women, including pregnancy	SSGO	2020
ES [13]	Executive summary of the diagnosis and treatment of urinary tract infection	SEIMC	2018
FR [14]	Practice guidelines for the management of adult community-acquired urinary tract infections	Joint report of SPILF and AFU	2014–update 2018
IE [15]	Management of urinary tract infections in pregnancy	Joint report of IOGRCPI/CSPDHSE	2015–update 2018
UK [16]	Urinary tract infection (lower): antimicrobial prescribing Pyelonephritis (acute): antimicrobial prescribing	NICE	2018
UK [17]	Pyelonephritis (acute): antimicrobial prescribing	NICE	2018
**South America**			
**Country/Region**	**Title**	**Organization**	**Year**
BR [21]	Recommendations for the clinical management of lower urinary tract infections in pregnant and non-pregnant women	Joint report of SBI/ FEBRASGO/SBU/SBPC/ML	2020
AR [22]	Argentine Intersociety Consensus on Urinary Infection	Joint report of SADI	2018–2019
CO [23]	Lower Urinary Tract Infections in Adults and Pregnant Women: A Consensus for Empirical Treatment	ACIN	2013
**Oceania**			
**Country/Region**	**Title**	**Organization**	**Year**
AU [22]	Perinatal Practice Guideline: Urinary Tract Infection in Pregnancy	SAH	2021

USA: United States of America; EU: Europe; DE: Germany; CH: Switzerland; ES: Spain; FR: France; UK: United Kingdom; IE: Ireland; BR: Brazil; AR: Argentina; CO: Colombia; AU: South Australia.

**Table 2 jcm-11-07226-t002:** Key points of frequently used antibiotics during pregnancy.

Antibiotics	Key Points (Guideline Reference)
Amoxicillin	Only if susceptible in the UC results [15,16] Good treatment option for GBSS [15,21] If <20 weeks of gestation and no alternative treatment is available [24]
Amoxicillin/clavulanate	Risk of necrotizing enterocolitis in neonates [15] Suitable throughout whole pregnancy and breastfeeding period [12]
Fosfomycin	Useful for patients with ESBLE [15] Not recommended if increased risk of premature birth [12]
Cephalexin, ceftriaxone and clindamycin	It can be used in mild penicillin allergy [15]
Cefuroxime	It can be used in mild penicillin allergy [15] Suitable throughout whole pregnancy and breastfeeding period [12]
Trimethoprim	Avoid in 1st T (folate antagonist) [7,11,14,16,24] If the only choice in the 1st T, use it with folic acid 5 mg/24 h [16] Suitable during breastfeeding period [12]
TMP/SMX	-Avoid in 1st T and 3rd T [7,12]
Nitrofurantoin	Avoid in 3rd T *: hemolysis in the newborn [11,15,16,21,22] Do not use in case of urine culture positive for certain species ** [23] Do not use if history of G6PD deficiency (risk of hemolysis) [15] Not suitable if patient has renal failure (GFR < 45 mL/min) [15,16]

* 3rd T: Third trimester of pregnancy (>36 weeks or sooner if early birth is planned); ** urine culture positive for: *M. morgannii*, *P. mirabilis*, *Providencia* spp., and *Serratia* spp.; 1st T: First trimester; UC: Urine culture; ESBLE: Extended-spectrum beta-lactamase producing *E. coli*; GBSB: Group B Streptococcal bacteriuria; G6PD: Glucose-6-phosphate dehydrogenase; GRF: Glomerular filtration rate.

**Table 3 jcm-11-07226-t003:** Antibiotic treatment for asymptomatic bacteriuria and cystitis during pregnancy.

	Asymptomatic Bacteriuria			Cystitis		
Antibiotic (Oral)	Treatment Line: Country/Region	Dosage (Guideline Reference)	Duration (Days)	Treatment Line: Country/Region	Dosage (Guideline Reference)	Duration (Days)
Nitrofurantoin	*1st Line:* UK, CO *5th Line:* FR *NS:* DE, IE, USA, AR, BR	100 mg/6–8 [19,20,21] 100 mg/12 h * [11,15,16,21] or 50 mg/6 h [11,16]	5–7 5–7 7	*1st Line:* UK, DE, CO *3rd Line:* FR *NS:* DE, IE, AR, BR	100 mg/6–8 [19,20,21] or 100 mg/12 h * [11,15,16] or 50 mg/6 h [11,16]	5–7 5–7 7
Amoxicillin	*1st Line:* FR *2nd Line:* UK *NS:* IE, AR, BR	500 mg/8 h [15,16,19] or 875 mg/12 h [19]	5 or 7 7	*1st Line:* UK *2nd Line:* AU *NS:* IE	500 mg/8 h [15,16,22]	5 or 7
Cephalexin (1stGC)	*2nd Line:* UK *NS:* IE, USA, AR, CO, BR	500 mg/6–8 h [19,20,21] or 500 mg/8–12 [15,16]	5–7	*1st Line:* AU *2nd Line:* UK *NS:* IE, AR, CO	500 mg/6–8 h [20,21] or 500 mg/8–12 h [15,16,22]	5–7
Pivmecillinam	*2nd Line:* FR *N.S:* DE	400 mg/8–12 h [11]	3 or 7	*2nd Line:* FR *NS:* DE	400 mg/8–12 h [11]	3 or 7
Fosfomycin	*1st Line:* CO *3rd Line*: FR *NS:* IE, DE, EU, CH, ES, BR	3 g [7,11,12,13,14,15,19,21]	Single dose	*1st Line:* FR, CO *NS:* IE, DE, EU, UK, ES, AR, BR	3 g [7,11,13,14,15,16,21]	Single dose
Trimethoprim	*4th Line:* FR	N.A	7	*1st Line:* AU	300 mg/24 h [22]	3
Amoxicillin/ clavulanate	*5th Line:* FR *NS:* IE, AR, CO	875 mg/12 h [20] or 500 mg/8 h [21] or 625 mg/8 h [15]	5–7	*1st Line:* CH *2nd Line:* AU *NS:* IE, CO, BR	500/125 mg/12 h [19,22] or 875/125 mg/12 h [19] or 500 mg/8 h [21] or 625 mg/8 h [12,15]	5–7
Cefixime (3rdGC)	*5th Line:* FR	N.A	7	*3rd Line:* FR	N.A	7
TMP/SMX	*5th Line:* FR *Last line:* DE *NS:* AR	800/160 mg/12 h [11,20]	4–7	*3rd Line:* CH *Last line:* DE	800/160 mg/12 h [11,12]	4–7
Cefuroxime (2ndGC)	*NS:* IE, BR	250–500 mg/12 h [19] 500 mg/12 h [12]	5–7	*2nd Line:* CH *NS:* IE, BR	250 mg/12 h [19] 500 mg/12 h [12,15]	5–7
Ampicillin/ sulbactam	*NS:* CO	1.5 g/12 h [21]	5–7			
Ampicillin	*NS:* USA	N.A	4–7			
Ciprofloxacin	-	-	-	*3rd Line:* FR	N.A	5–7

NS: Not specified; *: Macrocrystals/Prolonged release; 2ndGC: Second-generation cephalosporin; TMP/SMX: Trimethoprim/Sulfamethoxazole; EU: Europe; CO: Colombia; FR: France; UK: United Kingdom; DE: Germany; IE: Ireland; USA: United States of America; AR: Argentina; BR: Brazil; CH: Switzerland; ES: Spain; AU: Australia.

**Table 4 jcm-11-07226-t004:** Antibiotic acute pyelonephritis during pregnancy.

Antibiotic	Treatment Line: Country/Region	Dosage (Guideline Reference)	Duration (Days) *	Remarks (Guideline Reference)
Ceftriaxone IV	*1st Line:* IE, FR, AR, ES *2nd Line:* AU	1–2 g/24 h [14,15,22] 1–2 g/24 h [13,20]	10–14 or 7–10	If patient unstable: add G (i.e., es) [13,15] If contraindication for G [22] Consider 2 g dose in 2nd/3rd T of pregnancy [15]
Amoxicillin + Gentamicin IV	*1st Line:* AU, EU	Ax: 2 g/6 h [7,22] G: 5 mg/kg/24 h [7,22]	7–14	
Ampicillin + Gentamicin IV	*1st Line:* AU	Ap: 2 g/6 h [22] G: 5 mg/kg/24 h [22]	10–14	
2ndGC ± Aminoside IV	*1st Line:* EU	N.A	7–14	If sepsis: add aminoside [7]
Cephalexin (1stGC) PO	*1st Line:* UK	500 mg/8–12 h [17]	7–10	
Amoxicillin/Clavulanate PO	*1st Line:* CH	1 g/12 h or 625 mg/8 h [12]	5–7	
Cefuroxime (2ndGC)	*1st Line:* UK, ES *2nd Line:* CH	750 mg–1.5 g/6–8 h [17] or 500 mg/12 h [12]	7–10 3–5	If PO is not suitable [17] If sepsis: add aminoside [13]
Ciprofloxacin	*2nd Line:* FR	NA	7–14	
Clindamycin + Gentamicin IV	*2nd Line:* IE	C: 900 mg/8 h G: 1.5 mg/kg/8 h or 5 mg/kg/24 h [15]	10	If GBSS: choose vancomycin (1 g/12 h) or clindamycin based on the susceptible results [15]
Cefotaxime IV	*2nd Line:* AU	1 g/8 h [22]	10–14	If contraindication for G [22]
Gentamicin IV/IM	*2nd Line:* AR	240 mg/24 h [20]	10	If allergy to beta-lactams [20]
Aztreonam IV	*2nd Line:* AR, EU	1–2 g/8–12 h [20]	10	If allergy to beta-lactams [20]
TMP/SMX	3rd Line: CH	800/160 mg/12 h [12]	3–5	

EU: Europe; FR: France; UK: United Kingdom; DE: Germany; IE: Ireland; AR: Argentina; CH: Switzerland; ES: Spain; AU: Australia; G: Gentamicin; 2nd/3rd T: Second/Third trimester; *: days of parenteral + oral treatment.

## Data Availability

Data are available by contacting authors.

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
