# Peer review of "Which Antibiotic for Urinary Tract Infections in Pregnancy? A Literature Review of International Guidelines"

_jcm, 2022, doi:10.3390/jcm11237226_

Round 1

Reviewer 1 Report

Thank you for the opportunity of reviewing this article on antibiotic treatment of pregnancy-related UTI. It is a interesting study which may provede some valuable information for clinical practice.

As authors described in Discussion, "the lack of specifications of certain guidelines in terms of dosage and optimal duration of antibiotic courses in pregnancy made the results more general". So, I recommend to revise the manuscript so that readears can understand more clearly. There are some comments on this article.

1)Are there any recommendations for the adequate and/or inappropriate atibiotics fot the treatment at each stage (weeks) of gestation? Especially, what is the 1st choice treatment for the patients <16 weeks gestation?

2) Please add the flow diagram to show the recommended treatment stratagy based on the results of this study. In Discussion, I found some authors' explanations difficult to follow.

Author Response

All changes are in red

Reviewer: 1

Thank you for the opportunity of reviewing this article on antibiotic treatment of pregnancy-related UTI. It is a interesting study which may provede some valuable information for clinical practice.

Answer: We thank the reviewer for the positive comment about our manuscript.

As authors described in Discussion, "the lack of specifications of certain guidelines in terms of dosage and optimal duration of antibiotic courses in pregnancy made the results more general". So, I recommend to revise the manuscript so that readears can understand more clearly. There are some comments on this article.

Answer: We agree with the reviewer; however we mention that point on “Limitations” Line 316.

1)Are there any recommendations for adequate and/or inappropriate antibiotics for the treatment at each stage (weeks) of gestation? Especially, what is the 1st choice treatment for patients <16 weeks gestation?

Indeed, some antibiotics are suitable during the whole pregnancy, with precautions to consider following the guidelines selected, due to their risk to the fetus. It is also fundamental to tailor the treatment according to the organism sensitivities of the urine culture.

Certain antibiotics must be avoided during the first trimester, such as Trimethoprim and TMP/SMX; and others such as Nitrofurantoin must be avoided during the third trimester.

This information, according international guidelines, is summarized in Table 2, citing the most common used antibiotics during pregnancy.2) Please add the flow diagram to show the recommended treatment strategy based on the results of this study. In the Discussion, I found some authors' explanations difficult to follow.

2) Please add the flow diagram to show the recommended treatment stratagy based on the results of this study. In Discussion, I found some authors' explanations difficult to follow.

Answer: We agree with the reviewer. We have now added new information about the two scenarios described on our review, for better comprehension. It now reads –

“Based on those evidence we can describe a first scenario, screening for asymptomatic bacteriuria. All guidelines coming from North American, South American, the only one from Oceania and most European guidelines agreed on recommending systematic screening for ABU, even if most of the published studies have low-quality evidence. In summary, if we choose to do a screening for bacteriuria in pregnancy and we end up with a positive urine culture, then we can treat it tailoring the antibiotic treatment according to the weeks of pregnancy and urine culture sensitivity”.

And line 280:

“In summary this second scenario, which is treating a symptomatic UTI, recommends starting with an empiric antibiotic treatment according to the country's guidelines”.

Reviewer 2 Report

Congratulations on the present paper!

In the actual context, antibiotic treatment recommendations are essential.

The present paper is well-structured, concise, and comprehensive.

In my opinion, there should be added more recommendations from North America and Asian Continent.

Author Response

Answer: We thank the reviewer for the positive comment about our manuscript. We really appreciate it. 

We totally agree with you in that point. We have already collected the latest and available guidelines and recommendations from these continents. Unfortunately, a great amount of precise information was missing, for that reason they were excluded. We wanted to be more precise in our comments, so for that reason those guidelines were not included this time.